# An Inhibitor of the Sodium–Hydrogen Exchanger-1 (NHE-1), Amiloride, Reduced Zinc Accumulation and Hippocampal Neuronal Death after Ischemia

**DOI:** 10.3390/ijms21124232

**Published:** 2020-06-14

**Authors:** Beom Seok Kang, Bo Young Choi, A Ra Kho, Song Hee Lee, Dae Ki Hong, Jeong Hyun Jeong, Dong Hyeon Kang, Min Kyu Park, Sang Won Suh

**Affiliations:** 1Department of Physiology, College of Medicine, Hallym University, Chuncheon 24252, Korea; ttiger1993@gmail.com (B.S.K.); bychoi@hallym.ac.kr (B.Y.C.); rnlduadkfk136@hallym.ac.kr (A.R.K.); sshlee@hallym.ac.kr (S.H.L.); zxnm01220@gmail.com (D.K.H.); jd1422@hanmail.net (J.H.J.); bagmingyu50@gmail.com (M.K.P.); 2Department of Medical Science, College of Medicine, Hallym University, Chuncheon 24252, Korea; ehdgus6312@gmail.com

**Keywords:** global cerebral ischemia, amiloride, sodium–hydrogen exchanger-1, zinc, neuronal death, neuroprotection

## Abstract

Acidosis in the brain plays an important role in neuronal injury and is a common feature of several neurological diseases. It has been reported that the sodium–hydrogen exchanger-1 (NHE-1) is a key mediator of acidosis-induced neuronal injury. It modulates the concentration of intra- and extra-cellular sodium and hydrogen ions. During the ischemic state, excessive sodium ions enter neurons and inappropriately activate the sodium–calcium exchanger (NCX). Zinc can also enter neurons through voltage-gated calcium channels and NCX. Here, we tested the hypothesis that zinc enters the intracellular space through NCX and the subsequent zinc accumulation induces neuronal cell death after global cerebral ischemia (GCI). Thus, we conducted the present study to confirm whether inhibition of NHE-1 by amiloride attenuates zinc accumulation and subsequent hippocampus neuronal death following GCI. Mice were subjected to GCI by bilateral common carotid artery (BCCA) occlusion for 30 min, followed by restoration of blood flow and resuscitation. Amiloride (10 mg/kg, intraperitoneally (*i.p.*)) was immediately injected, which reduced zinc accumulation and neuronal death after GCI. Therefore, the present study demonstrates that amiloride attenuates GCI-induced neuronal injury, likely via the prevention of intracellular zinc accumulation. Consequently, we suggest that amiloride may have a high therapeutic potential for the prevention of GCI-induced neuronal death.

## 1. Introduction

Ischemic stroke is one of the most severe cerebral pathological conditions and can manifest via a number of clinical symptoms such as problems in cognition, dizziness, or loss of vision on one side of the visual field [1]. In particular, the development of ischemic conditions in the brain is very dangerous because only a momentary lack of adequate blood flow to the brain can lead to oxygen deprivation, insufficient nutrient provision, and potentially irreversible neural injury [2,3,4]. There are two basic types of ischemic injury: global and focal ischemia. Global ischemia impacts wide areas of brain tissue at once due to the blockage of blood flow to an entire region of the brain, while focal ischemia is limited to a specific region of the brain tissue and is due to the more local disruption of cerebral blood flow. Ischemia-induced brain damage can be recovered by early reperfusion, but this reperfusion process can also initiate independent cascades of cell death pathways such as zinc release, microglial activation, and blood–brain barrier (BBB) disruption [5]. If the interruption of blood circulation happens for an extended period prior to the restoration of circulation, brain damage can be permanent.

Furthermore, after a long period of ischemia, secondary brain damage occurs when blood flow is initiated, which is known as “reperfusion injury”. The sudden recovery of blood supply leads to mitochondrial dysfunction, producing excessive reactive oxygen species (ROS) and, finally, cell death [6]. Under healthy conditions, superoxide production via neuronal nicotinamide adenine dinucleotide phosphate (NADPH) oxidase is thought to play a part in normal physiological processes such as long-term potentiation and intracellular signaling [7,8]. However, under ischemic conditions, mitochondrial dysfunction and excessive ROS generation predominate and promote pathological responses such as leukocyte invasion and the disruption of the BBB [9,10]. Also, tissue pH is typically reduced to 6.0–6.5 during ischemia, falling even lower during severe ischemic conditions [11,12,13]. As a result, this study suggests that the increase in NADPH activity after ischemia may also promote cell death by acidifying brain cells.

Zinc is one of the most essential transition metals in our body, especially in the brain. Furthermore, zinc regulates physiological functions that control DNA synthesis, cell division, and signal transduction. Most of the zinc is present in a protein-bound form in neuronal cytoplasm. Free or chelatable zinc is localized within the vesicles of synaptic terminals [14,15]. Taken together, these lines of circumstantial evidence suggest that zinc plays a key role in maintaining cellular homeostasis. However, previous studies have demonstrated that excessive neuronal zinc accumulation occurs after traumatic brain injury, ischemia, hyperglycemia, and epilepsy [16,17,18,19]. Under these various pathological conditions, ROS changes the protein-mediated sequestration of zinc and, thus, increases intracellular free zinc levels which increases ROS production. If this condition is maintained for long periods, it can lead to neuronal death [18]. Several neurological injuries, accelerate zinc release from synaptic vesicles, and zinc then moves into neurons via multiple classes of membrane-bound channels [20,21].

Sodium–hydrogen exchangers (NHEs) are membrane transporters that mediate hydrogen efflux into cells. Sodium–hydrogen exchanger-1 (NHE-1) is a ubiquitous and essential membrane ion transporter that mediates the electroneutral exchange of hydrogen and sodium to regulate intracellular pH [22]. Other NHEs, namely, NHE-2–5, indicate more distinct cell-type- and tissue-dependent expressions than NHE-1 and play key roles in regulating transcellular sodium and potassium ion absorption [23]. The NHE-6 expression is localized to early endosomes [24]. The NHE-7 and -8 isoforms have not been localized in the brain, and the NHE-9 isoform is localized to late recycling endosomes [25]. Here, we focused on NHE-1 which is expressed in high abundance in the brain.

Amiloride, also known as thiazide, is related to other loop diuretic agents and blocks sodium hydrogen exchanger-1 [26]. Amiloride, an inhibitor of NHE-1, has demonstrated neuroprotective effects in various neuropathological conditions involving brain injuries such as ischemia, dementia, and epilepsy [27,28]. Previous studies have suggested that injection of amiloride can reduce cerebral-hypoxia-induced neuronal death after seizure and spinal cord injury [29,30,31]. Under the acidic conditions found in the post-ischemic state, as NHE-1 becomes activated, hydrogen is released into the extracellular space and sodium moves into the intracellular compartment. Thus, intracellular sodium is increased and the sodium–calcium exchanger (NCX) becomes activated, so that sodium is released into the extracellular space driving calcium entry, leading to a calcium overload state and, ultimately, resulting in neuronal death. Considering this, we hypothesized that not only calcium, but also zinc may enter via NCX and that NHE-1 blockade by amiloride decreases zinc accumulation and neuronal death through reduced NCX activity. Therefore, we investigated whether the administration of amiloride (10 mg/kg, intraperitoneally (*i.p.*)) reduces zinc accumulation, neuronal degeneration, oxidative stress, and microtubule damage after global cerebral ischemia. As a result, the present study proved that the administration of amiloride reduces ischemia-induced neuronal degeneration.

## 2. Results

### 2.1. Amiloride Reduced Global Cerebral Ischemia-Induced Hippocampus Neuronal Death after 24 Hour Post-Insult

To investigate whether amiloride has neuroprotective effects after global cerebral ischemia (GCI)-induced hippocampus neuronal death, experimental mice were immediately intraperitoneally injected with amiloride (10 mg/kg) after GCI. The mice were sacrificed 24 h after GCI with or without amiloride treatment. After insult, a histological evaluation to detect degenerating neurons was conducted in the hippocampal subiculum (Sub), cornus ammonis 1 (CA1), CA2, and CA3 regions. Fluoro-Jade B (FJB) staining showed widespread neuronal degeneration in the Sub, CA1, CA2, and CA3 regions of the hippocampus (*p* < 0.05) (Figure 1A). This staining protocol is a sensitive and selective marker of degenerating neurons. The number of degenerating neurons was increased in the GCI-induced group compared with the sham-operated group. When compared with the GCI-vehicle-treated groups, the amiloride-injected groups showed a dramatically reduced number of degenerating hippocampal neurons. Figure 1B shows the quantified FJB (+) neurons in the Sub, CA1, CA2, and CA3 regions. Amiloride-administered groups displayed a reduction of FJB (+) neurons of approximately 64% in the Sub (GCI-vehicle, 76.9 ± 8.4; GCI-amiloride, 27.2 ± 10.5), 84% in the CA1 (GCI-vehicle, 126.2 ± 18.5; GCI-amiloride, 19.5 ± 17.6), 83% in the CA2 (GCI-vehicle, 125.3 ± 22; GCI-amiloride, 20.7 ± 17.9), and 50% in the CA3 (GCI-vehicle, 220.6 ± 30.3; GCI-amiloride, 109.5 ± 29) regions compared with the vehicle-treated groups.

### 2.2. Amiloride Reduced Global Cerebral Ischemia-Induced Hippocampal Zinc Accumulation after 24 Hour Post-Insult

To estimate GCI-induced zinc accumulation, brain sections were histologically evaluated by *N*-(6-methoxy-8-quinolyl)-para-toluenesulfonamide (TSQ) staining 24 h after GCI. Zinc accumulation is known to advance the neuronal NADPH oxidase activity and ROS responsible for neuronal death. Under normal conditions, zinc levels are controlled by zinc transporters and zinc-binding proteins [32]. However, under conditions such as ischemia, traumatic brain injury, and seizure, neuronal death occurs in part by the destruction of zinc homeostasis. Thus, we performed TSQ fluorescence staining to confirm whether amiloride can reduce zinc accumulation in the brain hippocampal CA1 region. The intensity of TSQ staining in the CA1 region was reduced in the amiloride-treated group compared with the GCI-induced group (*p* < 0.05) (Figure 1C,D). Amiloride-administered groups displayed an approximately 44% reduction of TSQ (+) neurons in the CA1 region (GCI-vehicle, 36.9 ± 5.9; GCI-amiloride, 20.6 ± 4.1) compared with the vehicle-treated groups.

### 2.3. Amiloride Reduced Global Cerebral Ischemia-Induced Astrocyte and Microglial Activation 24 Hour Post-Insult

Multiple studies have previously reported that ischemia induces astrocyte and microglial activation in the brain hippocampal region [3,18,33], and it is well known that inflammation contributes to the severity of astrocyte and microglial activation. In addition, activated astrocytes are potentially harmful, because they can produce nitric oxide synthase (NOS) and neurotoxic nitric oxide (NO). To test whether amiloride affects GCI-induced glial activation, we performed immunofluorescence staining using the marker glial fibrillary acidic protein (GFAP) and ionized calcium-binding adaptor molecule 1 (Iba-1). We confirmed an assessment of the microglia number, morphology, and intensity and astrocyte intensity [34]. Global cerebral ischemia triggers astroglia and microglial activation which is considered to have macrophage-like activity. However, activated astrocytes were reduced by approximately 45% in the GCI–amiloride groups compared with the GCI-vehicle groups in the CA1 region (GCI-vehicle, 33.2 ± 3.3; GCI–amiloride, 18.2 ± 1.8). Further, activated microglia were reduced by 32% in the GCI–amiloride groups in the CA1 region (GCI-vehicle, 5.8 ± 0.3; GCI–amiloride, 3.9 ± 0.3) (Figure 2B,D).

### 2.4. Amiloride Reduces Global Cerebral Ischemia-Induced Oxidative Damage after 24 Hour Post-Insult

We estimated oxidative stress by using 4-hydroxynonenal (4HNE) staining. The brain was immunohistochemically stained with a 4HNE antibody 24 h after global cerebral ischemia induction to discover whether hippocampal neurons had experienced oxidative stress. The sham-vehicle and amiloride-injected groups showed no difference in 4HNE fluorescence signals. However, the 4HNE fluorescence signal of the GCI-vehicle groups increased in the hippocampal Sub, CA1, CA2, and CA3 regions. The amiloride-treated group showed a significant reduction of 4HNE intensity compared with the vehicle-treated groups (Figure 3A). Oxidative damage was reduced in the GCI-amiloride groups compared with the GCI-vehicle groups by approximately 42% in the Sub (GCI-vehicle, 21.8 ± 2.4; GCI-amiloride, 12.6 ± 0.5), 48% in the CA1 (GCI-vehicle, 23.5 ± 2.3; GCI-amiloride, 12.2 ± 1.5), 36% in the CA2 (GCI-vehicle, 21.5 ± 2.4; GCI-amiloride, 13.7 ± 1.4), and 48% in the CA3 (GCI-vehicle, 31.2 ± 3.3; GCI-amiloride, 16.1 ± 2.2) regions (Figure 3B).

### 2.5. Amiloride Reduced Global Cerebral Ischemia-Induced Microtubule Damage after 24 Hour Post-Insult

To evaluate whether GCI-induced microtubule damage occurred, brain sections were histologically processed with antibodies against microtubule-associated protein 2 (MAP2) 24 h after GCI. The GCI-vehicle groups showed a significant reduction in MAP2 immunoreactivity (IR) in the hippocampus and cortex compared with the GCI-amiloride group, indicating a loss of microtubules. Amiloride injection reduced GCI-induced microtubule damage compared with the vehicle-treated group (Figure 4A). Microtubule intensity was increased in the GCI-amiloride groups compared with the GCI-vehicle groups by approximately 42% in the Sub (GCI-vehicle, 41.9 ± 6.3; GCI-amiloride, 72.5 ± 7.2), 60% in the CA1 (GCI-vehicle, 43.3 ± 8.4; GCI-amiloride, 109.7 ± 12.5), 43% in the CA2 (GCI-vehicle, 56.2 ± 11; GCI-amiloride, 99.6 ± 10.1), and 59% in the CA3 (GCI-vehicle, 23.2 ± 5.9; GCI-amiloride, 57.2 ± 10.6) regions (Figure 4B).

### 2.6. Amiloride Prevented Global Cerebral Ischemia-Induced Blood–Brain Barrier Disruption after 24 Hour Post-Insult

To verify the degree of blood–brain barrier (BBB) disruption, we stained brain sections to evaluate extravasation of serum immunoglobulin G (IgG) by using immunohistochemistry as described before [35,36]. In sham-operated brain sections, leakage of IgG was not detected. However, in ischemia-induced mice, we observed excessive extravascular IgG leakage in the hippocampus (Figure 5A). Figure 5B shows a bar graph of the scale of IgG extravasation from the damaged BBB in the hippocampus. IgG leakage was reduced by 24% in the GCI-amiloride group compared with the GCI-vehicle group (GCI-vehicle, 1.37 ± 0.09; GCI-amiloride, 1.04 ± 0.07) (Figure 5B).

### 2.7. Amiloride Improves Global Cerebral Ischemia-Induced Survival of Hippocampal Neurons at 3 Days Post-Insult

To investigate whether amiloride promotes neuronal survival after GCI-induced hippocampus, amiloride (10 mg/kg) was immediately injected to the intraperitoneal space after termination of the blood reperfusion process. Both vehicle- and amiloride-administrated cohorts were sacrificed at 3 days following ischemic insult. After the insult, a histological evaluation using NeuN staining to detect and quantify newly generated neurons was conducted in the hippocampal Sub, CA1, CA2, and CA3 regions. NeuN positive neurons were widespread in the hippocampal regions we examined (*p* < 0.05) and increased in number rapidly (Figure 6A). Figure 6B shows the counted NeuN (+) neurons in the hippocampal regions. Amiloride-administered groups displayed an increase of NeuN (+) neurons of 26% in the Sub (GCI-vehicle, 115.1 ± 4.0; GCI-amiloride, 156.4 ± 10.8), 17% in the CA1 (GCI-vehicle, 177.7 ± 3.5; GCI-amiloride, 213.4 ± 11.1), 17% in the CA2 (GCI-vehicle, 219.5 ± 5.9; GCI-amiloride, 265 ± 10.9), and 17% in the CA3 (GCI-vehicle, 251.2 ± 8.2; GCI-amiloride, 304.5 ± 13.2) regions compared with the vehicle-treated groups.

## 3. Discussion

Previous studies have demonstrated that amiloride showed neuroprotective effects in two different stroke models; middle cerebral artery occlusion (MCAO) in rat and transient forebrain ischemia in gerbil [27,37]. However, no studies have been performed with the global cerebral ischemia model with mice. The present study investigated whether amiloride administration has potential therapeutic effects for GCI-induced hippocampus neuronal damage and zinc accumulation via inhibition of NHE-1 in mice. Consequently, we found that amiloride significantly reduced zinc accumulation, neuronal degeneration, oxidative damage, microtubule damage, astrocyte and microglial activation, and BBB disruption.

Under ischemic conditions, blood flow to the brain is rapidly and dramatically reduced. This phenomenon leads to a lack of oxygen and other substrates to the nervous tissue. Thus, the extracellular concentration of glucose is quickly reduced [38,39,40]. As a result, the physiological glycolysis process was disturbed, and glucose 6-phosphate altered ribulose 5-phosphate by the pentose phosphate pathway (PPP). While glucose is modified to ribulose 5-phosphate, hydrogen is released and triggers NHE-1 [41,42]. When the intracellular hydrogen ion concentration increases, NHE-1 is activated in the cell membrane. NHE-1 activation contributes to neuronal electron exchange via sodium and hydrogen ion exchange across the cell membrane [43]. Previous studies have shown that NHE-1 inhibitors reduced the activity of NCX [27,44,45,46,47]. Furthermore, additional studies demonstrated that zinc enters intracellularly through NCX [48,49]. Thus, we can speculate that NHE-1 inhibition may indirectly inhibit zinc influx into neurons. Also, ischemia-induced ROS formation degrades zinc-binding proteins in the intracellular space, thus significantly increasing free zinc levels within the intracellular space [18]. As this cycle becomes sustained, the accumulation of excessive amounts of zinc contributes to neuronal cell death [3,18,27,50] (Figure 7).

Amiloride has previously been described as a diuretic and as a non-specific inhibitor for NHE-1. However, it has been used as an NHE-1 inhibitor in several studies [26,51]. Additionally, amiloride has been known to cross the blood–brain barrier and has neuroprotective effects against global cerebral ischemia. Several studies have demonstrated that amiloride has neuroprotective properties using a dose of 10 mg/kg [27,52,53,54]. Following this logic, we hypothesized that GCI-induced hippocampal damage can be protected by reducing intracellular zinc accumulation through inhibition of NHE-1 [27,55].

The histological evaluation, we performed further supports our hypothesis. The number of hippocampal degenerating neurons was estimated using FJB staining. The number of FJB fluorescence-signal-positive neurons in the hippocampal Sub, CA1, CA2, and CA3 regions was significantly reduced in the amiloride-administered group. Intraneuronal free zinc accumulation was displayed by TSQ staining. The number of the TSQ-positive neurons were significantly reduced in the amiloride-administered groups compared with the vehicle groups. These results demonstrated that the intraneuronal free zinc accumulation was reduced by inhibiting NHE-1 channels using amiloride. We confirmed that under ischemic conditions, zinc accumulation increases, and neuronal death occur. However, blocking NHE-1 by injection of amiloride reduces zinc accumulation and, subsequently, neuronal death was reduced. Additionally, we confirmed the presence of live neurons at 3 days following ischemic insult and stained for NeuN to identify neurons that survived the insult. The number of NeuN positive neurons in the hippocampal Sub, CA1, CA2 and CA3 regions were greater in the amiloride-administered group. So, we concluded that the administration of amiloride improved the survival of hippocampal neurons after 3 days post-insult.

Astrocytes and microglia play important roles in the brain. During neurological disorders such as ischemia, neuroinflammation, and neurodegenerative disease, astrocytes and microglia are over activated. Under ischemic conditions, activated astrocytes, together with reactive microglia, release several pro-inflammatory factors such as tumor necrosis factor-α (TNF-α), ROS, NO, and interleukin-1β (IL-1β), which exacerbate tissue damage [18,56,57,58]. Previous studies have demonstrated that NHE-1 expression occurs in astrocyte and microglial cells. In addition, NHE-1 activity caused by ischemia activates astrocyte and microglial cells, leading to neuronal death [18,59]. So, we thought that the administration of amiloride, an NHE-1 inhibitor, would reduce the activation of astrocytes and microglia, and thus reduce neuronal death. The present study verified astrocyte activation by GFAP and microglia activation by Iba-1 immunofluorescence staining in the hippocampal CA1 region. We found that amiloride administration reduced reactive astrocyte and microglial activation after GCI. 

Ischemic damage leaded to microtubule damage and ROS production. In the present study, we found that amiloride administration reduced microtubule damage after GCI. ROS formation is caused via multiple intracellular signaling cascades, such as iron-associated free radical formation, depletion of antioxidant enzymes, and an increase in the breakdown of lipids and fatty acids after GCI [60]. Because of this, the zinc accumulation described above occurs and microtubules are damaged [18,61]. In addition, several previous studies have suggested that peroxynitrite (PN) toxicity is mediated by intracellular zinc release [62,63]. Peroxynitrite is produced by a combination of nitric oxide and superoxide. It has been reported that PN are endogenous reactive nitrogen species formed when superoxide radicals, or oxygen reacts with nitric oxide formed by inducible nitric oxide synthase (iNOS). Peroxynitrite can induce cytoplasmic free zinc release, mitochondria dysfunction, and lead to BBB disruption, and finally neuronal death, in several types of brain injuries [64,65,66]. To test whether ROS activation was reduced by amiloride, 4HNE staining was performed in the hippocampal Sub, CA1, CA2, and CA3 regions. The 4HNE fluorescence signal was significantly increased in the GCI-vehicle group. For the amiloride-administered group after GCI, the 4HNE fluorescence signal was significantly decreased in hippocampal regions compared with the GCI-vehicle group. Amiloride, which inhibits NHE-1 and decreases NCX activity, reduces ROS formation by regulating the intracellular ion balance.

Finally, we evaluated the BBB balance after GCI. Abnormal intracellular zinc accumulation and vesicular zinc release may mediate BBB disruption after brain insults such as ischemia, multiple sclerosis, and traumatic brain injury [67]. Thus, administration of amiloride may decrease zinc accumulation and reduces BBB disruption after GCI. After GCI insult, the BBB was destroyed, leading to extravasation of plasma components, such as erythrocytes, leukocytes, and several immunoglobulins. BBB disruption triggers neurodegenerative processes and produces neurotoxic substrates, which causes brain dysfunction that is deleterious to synapse function and disturbs neural transmission [68,69]. So, to evaluate the effects of amiloride on BBB disruption, we conducted IgG staining. As a result, we found that the IgG staining intensity was reduced in the amiloride-administered group compared with the vehicle group after GCI. This result indicates that the neuroprotective provided by amiloride administration might be mediated via protection against BBB disruption, which is one of the main mechanisms associated with GCI-induced neuronal death [70].

Taken together, we conclude that inhibition of NHE-1 by amiloride reduces neuronal death and zinc accumulation. We found that amiloride reduced various deleterious features associated with GCI, such as neurodegeneration, zinc accumulation, oxidative damage, microtubule damage, glial activation, and BBB disruption, which strongly indicates that amiloride administration has neuroprotective effects by decreasing ROS production and zinc accumulation in hippocampal neurons through the inhibition of NHE-1. Therefore, the present study suggests that amiloride can be a potential therapeutic tool to prevent ischemia-induced neuronal death.

## 4. Materials and Methods 

### 4.1. Ethics Statement

The present study was performed in accordance with the protocols of the Guidelines for the Use and Care of Laboratory Animals, allowed by the National Institutes of Health. Animal studies were conducted in accordance with the guidelines of the Committee on Animal Use for Study and Education at Hallym University (protocol # Hallym-2018-32; Data of approval: July 19,2018). We sacrificed mice under isoflurane anesthesia to minimize any pain and suffering.

### 4.2. Experimental Animals

The present study used 2–3 month old adult male C57BL/6J mice (20–25 g, DBL Co., Chungcheongbuk-do, Eumseong-gun, Korea). The mice were housed three to four mice per cage under conditions of sustained temperature (20 ± 2 °C) and humidity (55% ± 5%). Animal room lights were managed automatically, turned on and off in a 12 h cycle (on at 6:00 a.m. and off at 6:00 p.m.).

### 4.3. Global Cerebral Ischemia Surgery

Male C57BL/6J mice (aged 2–3 months, weight 20–25 g) from DBL (Chungcheongbuk-do, Eumseong-gun, Korea) were used as controls. The mice were anesthetized with 2% isoflurane in a 75:25 mixture of oxygen and nitrous oxide. Core temperature was maintained at 36.7–37.5 °C with a homeothermic blanket control unit (Harvard Apparatus, Holliston, MA, USA). Bilateral common carotid arteries (BCCAs) were exposed through a midline neck incision. The BCCAs were loosely encircled with a 4/O silk suture before the occlusion. Aneurysmal clips were used to occlude BCCAs. Mice were subjected to common carotid artery occlusion for 30 min while anesthetized with 1% isoflurane [71,72,73]. The aneurysmal clips were removed, and the BCCAs were inspected for normal recovery of blood flow after the end of the 30 min ischemic period. Anesthetics were discontinued following the suture of the skin incision. When mice confirmed spontaneous respiration, they were returned to a recovery room retained at 37 °C. Sham-operated animals received the same neck skin incision under isoflurane anesthesia without BCCA occlusion.

### 4.4. Amiloride Administration

To confirm the effect of amiloride on GCI-induced neuronal death, the experimental groups were divided into four groups: sham (vehicle, amiloride) and global cerebral ischemia (vehicle, amiloride). The amiloride-treated groups were administrated amiloride (10 mg/kg, i.p.) dissolved in 0.9% normal saline. After GCI induction, we immediately injected the amiloride into the intraperitoneal space. The vehicle group was given 0.9% normal saline instead of amiloride. All experimental groups were sacrificed 24 h after GCI.

### 4.5. Brain Sample Preparation

Mice were sacrificed at 24 h or 3 days after GCI using urethane (1.5 g/kg, i.p.) to deeply anesthetize them. After anesthetizing, mice were perfused transcardially with 0.9% saline, followed by 4% paraformaldehyde (PFA). Afterwards, harvested brains were post-fixed for approximately 1 h in 4% PFA. After fixation in PFA, brains were moved into a 30% sucrose solution overnight for cryoprotection. After the brain sank to the bottom of the sucrose solution, they were frozen in the freezing medium for 10 min and then cut with cryostats at 30 μm thicknesses. Brain slices were kept in storage solution until used for immunohistochemistry and immunofluorescence staining.

### 4.6. Confirmation of Hippocampal Neuronal Death

To confirm neuronal death after GCI, brain sections (30 μm) were put on gelatin-coated slides (Fisher Scientific, Pittsburgh, PA, USA). To detect degenerating neurons, brain slices were stained by the FJB staining method [74,75]. Firstly, a slide with a brain section was soaked in a 100% ethanol solution for 3 min, a 70% ethanol solution for 1 min, distilled water for 1 min, and then in 0.06% potassium permanganate for 15 min. Next, the slides were put into 0.001% FJB (Histo-Chem Inc., Jefferson, AR, USA) solution for 30 min and washed three times for 10 min in distilled water. After washing, slides were dried by a gentle air flow (Labtech, Co., Ltd., Namyangju, Korea), dehydrated in xylene for 2 min, and then mounted with DPX (Sigma-Aldrich Co., St. Louis, MO, USA). Slides were checked under a fluorescence microscope (Olympus, Japan) via blue (450–490 nm) excitation light. We chose about six to eight coronal brain sections that were collected from each mouse. A blinded observer counted the FJB-positive cells. The FJB-positive cells were counted and evaluated in the hippocampal Sub, CA1, CA2, and CA3 regions from the bilateral hemisphere. The total number of FJB-positive cells from the hippocampal region was used for statistical analysis. A blinded observer counted the FJB-positive cells. The FJB-positive cells were counted and evaluated in the hippocampal Sub, CA1, CA2, and CA3 regions from the bilateral hemisphere. The total number of FJB-positive cells from the hippocampal region was used for statistical analysis.

### 4.7. Confirmation of Hippocampal Zinc Translocation

Intracellular free zinc was verified using TSQ staining [76]. Mice were sacrificed 24 h after amiloride (10 mg/kg, i.p.) administration and the fresh frozen, but not fixed, brains were coronally sectioned at 10 μm thicknesses in a −15 °C cryostat, then mounted on gelatin-coated slides and dried. Five evenly spaced sections were chosen from the hippocampal region of each brain and soaked in a solution of 4.5 mmol/L TSQ (Enzo Life Science, Enzo Biochem, Inc, Farmingdale, New York, NY, USA, ENZ-52153) for 1 min, then washed for 1 min in 0.9% saline. Each sample was photographed with a microscope under 360 nm UV light and a 500 nm long-pass filter. We used the Image J (National Institute of Health, Bethesda, Rockville, MD, USA) program to measure zinc intensity and evaluated the mean gray value.

### 4.8. Evaluation of Hippocampal Oxidative Stress

To analyze oxidative damage induced by the lipid peroxidation product from the brain sections, 4HNE was detected by immunofluorescence staining. 4HNE antibodies (Alpha Diagnostic Intl. Inc., San Antonio, TX, USA) for immunohistochemical staining were used as in previous studies [77,78]. Brain sections were soaked in a monoclonal mouse anti-4HNE serum (diluted 1:500, Alpha Diagnostic Intl. Inc., San Antonio, TX, USA) with the PBS containing 0.3% TritonX-100 overnight in a 4 °C incubator. After overnight incubation, brain sections were washed three times for 10 min with 0.01 M PBS, and then the brain sections were also soaked in a solution of Alexa-Fluor-594-conjugated donkey anti-mouse IgG secondary antibody (diluted 1:250, Invitrogen, Grand Island, NY, USA) for 2 h at room temperature (RT). The brain sections were raised on gelatin-coated slides for analysis under a microscope. We used the Image J (NIH, Bethesda, Rockville, MD, USA) program to measure the oxidative injury and measured the mean gray value [79].

### 4.9. Evaluation of Hippocampal Microtubule Damage

To analyze microtubule damage from the brain sections, MAP2 was detected by immunofluorescence staining. MAP2 antibodies (Alpha Diagnostic Intl. Inc., San Antonio, TX, USA) for immunohistochemical staining were used as in a previous study [80]. Brain sections were soaked in a polyclonal rabbit anti-MAP2 serum (diluted 1:200, Alpha Diagnostic Intl. Inc., San Antonio, TX, USA) with PBS containing 0.3% TritonX-100 overnight in a 4 °C incubator. After overnight incubation, we washed the sections three times for 10 min with 0.01 M PBS, and then the brain sections were soaked in a solution of Alexa-Fluor-488-conjugated donkey anti-rabbit IgG secondary antibody (diluted 1:250, Invitrogen, Grand Island, NY, USA) for 2 h at RT. The brain sections were raised on gelatin-coated slides for analysis under a microscope. We used the Image J (NIH, Bethesda, Rockville, MD, USA) program to measure the microtubule damage and measured the mean gray value.

### 4.10. Evaluation of Hippocampal Astrocytes and Microglia

To analyze astrocyte and microglial activation, we performed Iba-1 and GFAP staining. Staining was used with a mixture of goat antibody to mouse Iba-1 (diluted 1:500, Abcam, Cambridge, UK) and rabbit antibody to mouse GFAP (diluted 1:1000, Abcam, Cambridge, UK). Following incubation in 0.01 M PBS containing 0.3% TritonX-100, we left it overnight in a 4 °C incubator. After overnight incubation, we washed the sections three times for 10 min with 0.01 M PBS. Then, the sections were soaked in a secondary antibody (Alexa-Fluor-488-conjugated donkey anti-rabbit IgG secondary antibody and Alexa-Fluor-594-conjugated donkey anti-goat IgG secondary antibody, both diluted 1:250, Invitrogen, Grand Island, NY, USA) for 2 h at RT. The brain sections were raised on gelatin-coated slides for analysis under a microscope. We used the Image J (NIH, Bethesda, Rockville, MD, USA) program to measure the astrocytes. In the case of microglia, five brain sections were scored with the same area (20× magnification) of the hippocampal CA1 region. The functional standards of microglia cells were their number, morphology, and intensity of microglia activation. Iba-1-immunoreactive cell score of 0: no cells are present; 1:1–9 cells; 2:10–20 cells; and 3:>20 cells with continuous processers per 100 μm^2^. Morphology score of 0:no activated morphology (amoeboid morphology with enlarged soma and thickened processes); 1:1–45% of microglia activation; 2:45–90% of microglia activation; and 3: >90% of microglia with the activated morphology. The intensity of microglial activation was measured using the Image J (NIH, Bethesda, Rockville, MD, USA) program. After the measurements, an intensity score of 1:0–19% expression; 2:20–29% expression; and 3:>29% expression. Therefore, the total score summed up the three scores depending on the categories, ranging from 0 to 9 [18,81,82].

### 4.11. Evaluation of BBB Disruption

To analyze the putative breakdown of the BBB, we used immunohistochemistry to find serum IgG leakage [83]. To detect IgG-like immunoreactivity, the ABC immunoperoxidase protocol was used [84]. Mouse brains were fixed by transcardiac perfusion with 0.9% normal saline, followed by 4% paraformaldehyde. We used anti-mouse IgG (diluted 1:250, Burlingame, Vector, CA, USA) which can discover leakages of IgG when the BBB is damaged. After washing in 0.01 M PBS, brain sections were deeply soaked in the ABC complex mixture (Vector, Burlingame, CA, USA) for 2 h at RT. The immunoreactivity was visualized with 0.06% 3,3’-diaminobenzidine (DAB ager, Sigma–Aldrich Co., St. Louis, MO, USA) in 0.1 M PBS buffer. Leaked IgG extravasations were detected using a bright-field microscope.

### 4.12. Evaluation of Live Hippocampal Neurons

To assess the number of live neurons present in a sample, NeuN was detected by immunofluorescence staining. NeuN antibodies (diluted 1:500, EMD Millipore, Billerica, MA, USA) for immunohistochemical staining were used as in previous studies. Brain sections were soaked in a monoclonal rabbit anti-NeuN serum with PBS containing 0.3% TritonX-100 overnight in a 4 °C incubator. After overnight incubation, brain sections were washed three times for 10 min each with 0.01 M PBS, and then the brain sections were also soaked in a solution of Alexa-Fluor-594-conjugated donkey anti-rabbit IgG secondary antibody (diluted 1:250, Invitrogen, Grand Island, NY, USA) for 2 h at room temperature (RT). The brain sections were raised on gelatin-coated slides for analysis under a microscope. A blinded observer counted the NeuN-positive cells. NeuN-positive cells were counted and evaluated in the hippocampal Sub, CA1, CA2, and CA3 regions from each hemisphere. The total number of NeuN-positive cells from the hippocampal region was used for statistical analysis [18].

### 4.13. Statistical Analysis

We conducted nonparametric testing to confirm any statistical significance between the experimental groups. Data were analyzed using the Bonferroni post-hoc test, and the Kruskal–Wallis test was employed to compare among the four groups. For comparison across two groups, data were analyzed using the Mann–Whitey U test. Data are displayed as the mean ± SEM. Statistical significance is described as *p* < 0.05.

## 5. Conclusions

This present study supports the hypothesis that the administration of amiloride protects against hippocampal neuronal death and zinc accumulation after GCI. In addition, inhibition of NHE-1 by amiloride may have considerable therapeutic potential for the prevention of GCI.

## Figures and Tables

**Figure 1 ijms-21-04232-f001:**
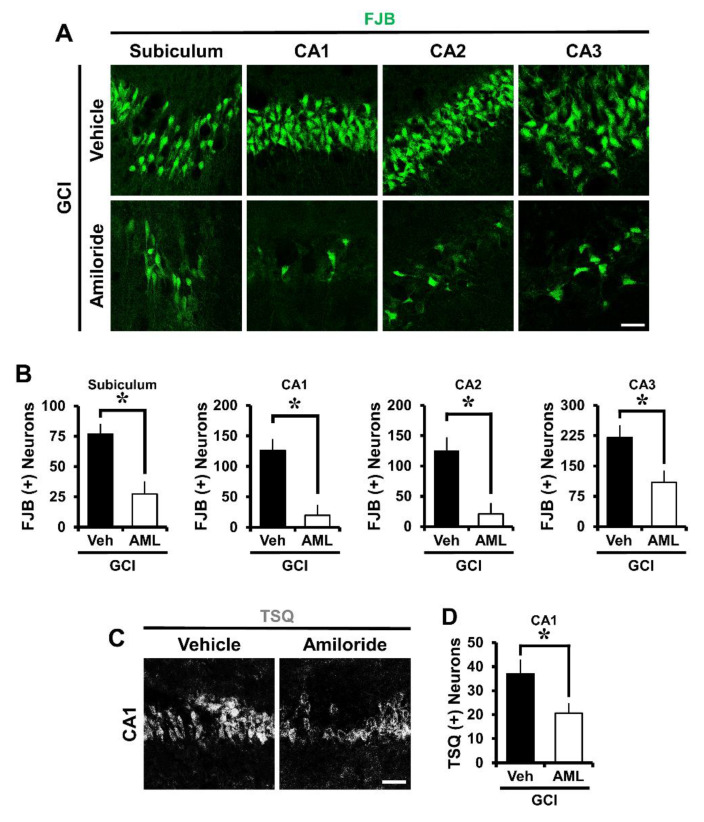
Amiloride treatment decreased the number of degenerating neurons and zinc accumulation after global cerebral ischemia (GCI). GCI-induced hippocampus neuronal death was confirmed in the subiculum (Sub), cornus ammonis 1 (CA1), CA2, and CA3 regions after ischemic insult. Zinc accumulation was confirmed in the CA1 region after ischemic insult. (**A**) Fluorescent images show degenerated neurons in the Sub, CA1, CA2, and CA3 regions. Intraperitoneal post-treatment with amiloride (10 mg/kg) reduced neuronal death in the Sub, CA1, CA2, and CA3 regions at 24 h after ischemia. Scale bar = 20 μm. (**B**) Bar graph displaying the quantification of degenerating neurons in the hippocampal regions. The number of FJB (+) neurons was decreased in the amiloride-injected (10 mg/kg) group in the Sub, CA1, CA2, and CA3 regions compared with the vehicle-treated group (GCI-vehicle, *n* = 8; GCI-amiloride, *n* = 8). (**C**) Representative images show *N*-(6-methoxy-8-quinolyl)-para-toluenesulfonamide (TSQ) (+) neurons in the CA1 region. Scale bar = 20 μm. (**D**) The bar graph indicates the TSQ (+) neurons in the hippocampal CA1 region (GCI-vehicle, *n* = 8; GCI-amiloride, *n* = 10). Data are mean ± S.E.M. * Considerably different from the vehicle-treated group, *p* < 0.05. (Mann–Whitney U test (B) Sub: z = 2.626, *p* = 0.007; CA1: z = 2.838, *p* = 0.003; CA2: z = 2.836, *p* = 0.003; CA3: z = 2.205, *p* = 0.028; (D) CA1: z = 2.134, *p* = 0.034).

**Figure 2 ijms-21-04232-f002:**
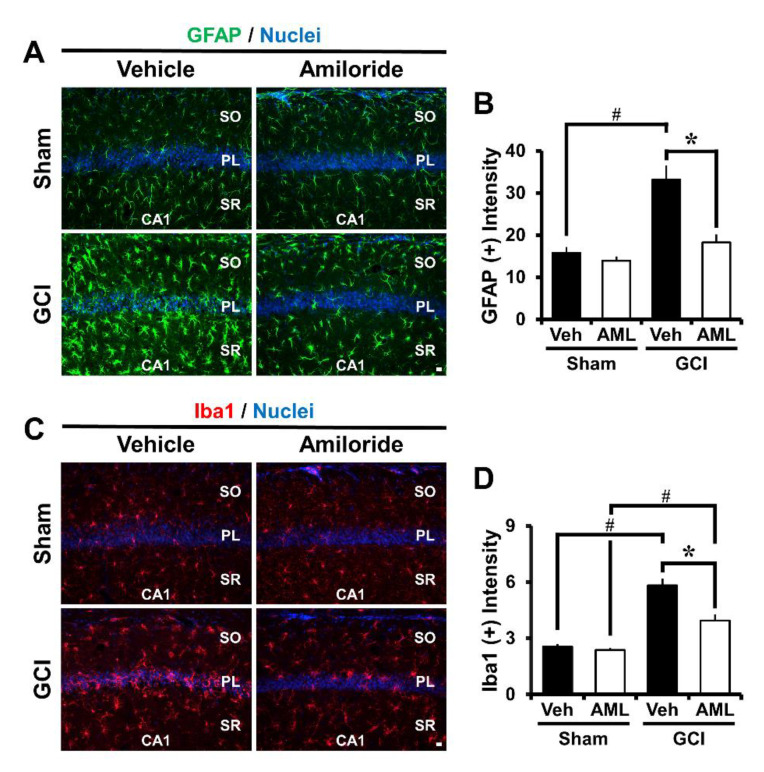
Amiloride treatment reduced astrocyte and microglial activation. GCI induces an inflammatory response by promoting astrocyte and microglia activation in the damaged brain. This figure shows astrocyte and microglia activation in the CA1 region 24 h after GCI. (**A**,**C**) show glial fibrillary acidic protein (GFAP) and Ionized calcium-binding adaptor molecule-1 (Iba-1) activation and (**B**,**D**) show GFAP and Iba-1 quantification in the CA1 region from sham-operated or GCI-induced mice. It was increased in the GCI-induced group compared with the sham-operated group. However, the amiloride-treated group showed reduced astrocyte and microglia activation after GCI. Scale bar = 20 μm. (Sham-vehicle, *n* = 5; sham-amiloride, *n* = 5; GCI-vehicle, *n* = 6; GCI-amiloride, *n* = 8). Data are mean ± SEM. * Considerably different from the vehicle-treated group, *p* < 0.05; # sham versus vehicle-operated group; sham versus vehicle-treated group, *p* < 0.05. (Kruskal–Wallis test (B) Chi square = 14.612, df = 3, *p* = 0.002; (D) Chi square = 18.606, df = 3, *p* < 0.001) (SO: stratum oriens; PL: pyramidal cell layer; SR: stratum lacunosum-moleculare).

**Figure 3 ijms-21-04232-f003:**
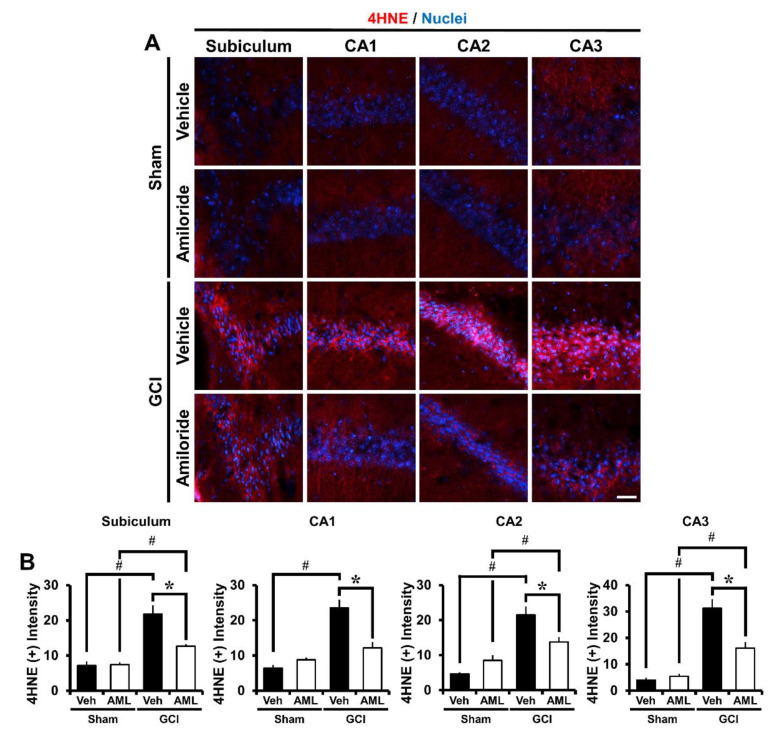
Amiloride reduced oxidative injury after GCI. Oxidative injury was detected by 4-hydroxynonenal (4HNE, red color) staining from the hippocampal Sub, CA1, CA2, and CA3 regions 24 h after GCI. (**A**) Sham-operated groups showed minimal 4HNE fluorescence signals in the hippocampus. Amiloride-treated groups showed reduced immunoreactive fluorescence intensity for 4HNE in the hippocampus compared with the vehicle-treated group after GCI. Scale bar = 20 μm. (**B**) The bar graph presents the 4HNE fluorescence intensity in the Sub, CA1, CA2, and CA3 regions. The fluorescence intensity showed a significant difference among groups (sham-vehicle, *n* = 6; sham-amiloride, *n* = 5; GCI-vehicle, *n* = 8; GCI-amiloride, *n* = 8). Data are mean ± S.E.M. * Considerably different from the vehicle-treated group, *p* < 0.05; # sham versus vehicle-operated group, sham versus vehicle-treated group, *p* < 0.05. (Kruskal–Wallis test (B) Sub: Chi square = 22.444, df = 3, *p* < 0.001; CA1: Chi square = 17.896, df = 3, *p* < 0.001; CA2: Chi square = 20.967, df = 3, *p* < 0.001; CA3: Chi square = 20.986, df = 3, *p* < 0.001).

**Figure 4 ijms-21-04232-f004:**
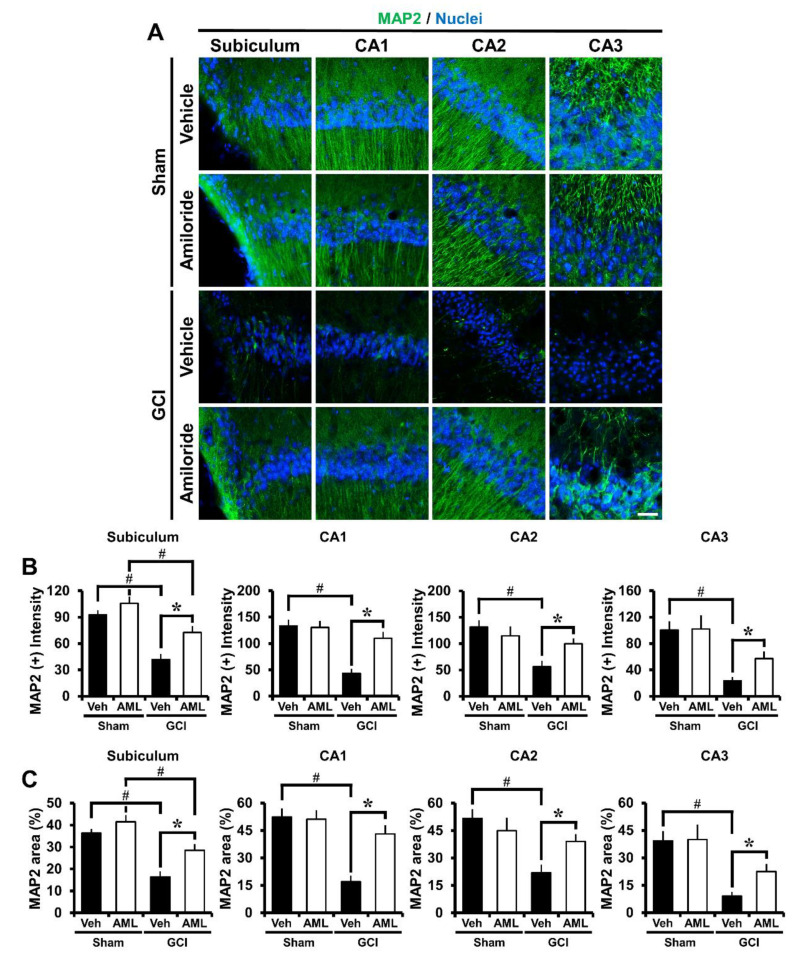
Microtubule damage was detected by microtubule-associated protein 2 (MAP2, green color) staining at the hippocampal Sub, CA1, CA2, and CA3 regions 24 h after GCI. (**A**) Sham-operated groups showed MAP2 fluorescence signals in the hippocampus. Amiloride-administered groups showed a reduced microtubule loss in the hippocampal regions compared with the vehicle-treated group. Scale bar = 20 μm. (**B**) The bar graph indicates the MAP2 fluorescence intensity in the hippocampus. (**C**) The bar graph indicates the MAP2 percent area in the hippocampus (sham-vehicle, *n* = 6; sham-amiloride, *n* = 5; GCI-vehicle, *n* = 8; GCI-amiloride, *n* = 8). Data are mean ± S.E.M. * Considerably different from the vehicle-treated group, *p* < 0.05; # sham versus vehicle-operated group, sham versus vehicle-treated group, *p* < 0.05. (Kruskal–Wallis test (B) Sub: Chi square = 18.901, df = 3, *p* < 0.001; CA1: Chi square = 15.166, df = 3, *p* < 0.002; CA2: Chi square = 15.054, df = 3, *p* < 0.002; CA3: Chi square = 17.137, df = 3, *p* < 0.001).

**Figure 5 ijms-21-04232-f005:**
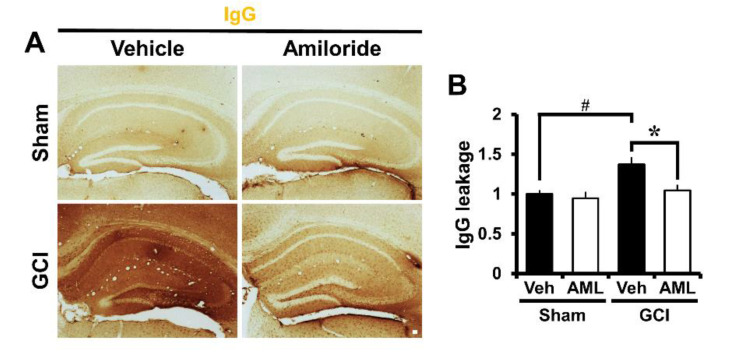
GCI-induced blood–brain barrier (BBB) disruption was decreased by amiloride administration. Brain sections were stained with antibodies against IgG to detect BBB disruption. (**A**) Indicates magnification (4×) of a microscopic image of IgG staining in the hippocampus in each group. These images indicate that BBB disruption occurred after GCI. The GCI-amiloride group had decreased leakage of serum IgG in the hippocampus compared with the GCI-vehicle group. Scale bar = 100 μm. (**B**) Bar graph shows the quantification of IgG serum extravasation in the hippocampus (sham-to-GCI ratio, sham-vehicle, *n* = 6; sham-amiloride, *n* = 5; GCI-vehicle, *n* = 7; GCI-amiloride, *n* = 8). Data are mean ± SEM. * Considerably different from the vehicle-treated group, *p* < 0.05; # sham versus vehicle-operated group; *p* < 0.05. (Kruskal–Wallis test (B) Chi square = 11.126, df = 3, *p* = 0.011).

**Figure 6 ijms-21-04232-f006:**
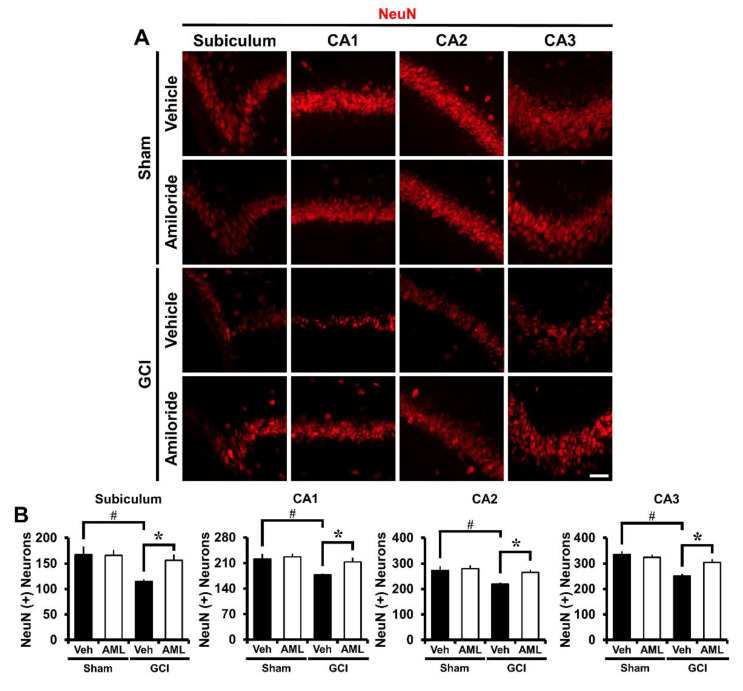
Amiloride treatment improves neuronal survival after GCI. The presence of live neurons after ischemic insult was confirmed in the Sub, CA1, CA2, and CA3 regions. (**A**) Fluorescent images show surviving neurons in the Sub, CA1, CA2, and CA3 regions. Intraperitoneal post-treatment with amiloride (10 mg/kg) increased the number of live neurons in the Sub, CA1, CA2, and CA3 regions at 3 days post-ischemia. Scale bar = 20 μm. (**B**) Bar graph displaying the quantification of surviving neurons in hippocampal subregions. The number of NeuN (+) neurons was increased in the amiloride-injected (10 mg/kg) group in the Sub, CA1, CA2, and CA3 regions compared with the vehicle-treated group (GCI-vehicle, *n* = 8; GCI-amiloride, *n* = 6). Data are mean ± SEM. * Considerably different from the vehicle-treated group, *p* < 0.05; # sham versus vehicle-operated group, *p* < 0.05. (Kruskal–Wallis test (B) Sub: Chi square = 14.214, df = 3, *p* = 0.003; CA1: Chi square = 13.422, df = 3, *p* = 0.004; CA2: Chi square = 14.249, df = 3, *p* = 0.003; CA3: Chi square = 16.158, df = 3, *p* = 0.001).

**Figure 7 ijms-21-04232-f007:**
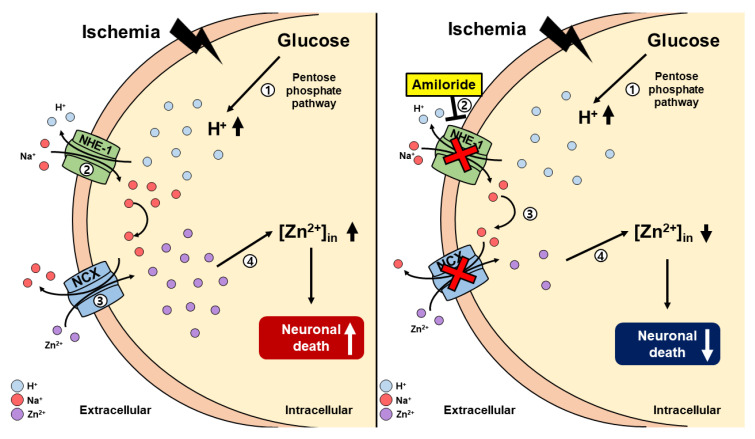
This schematic illustration shows amiloride action via inhibition and downregulation of the sodium–hydrogen exchanger-1 (NHE-1) channel. (**A**) [1] Global cerebral ischemia insult results in increasing levels of intracellular hydrogen. [2] Hydrogen ions are moved to the extracellular space via NHE-1. Sodium ions are moved to the intracellular space via NHE-1. [3] When sodium is overloaded in the intracellular space, it is released into the extracellular space through the sodium–calcium exchanger (NCX). Extracellular zinc enters the cell through NCX. [4] Intracellular zinc accumulation occurs, leading to neuronal death. (**B**) However, [1] after global cerebral ischemia, [2] amiloride administration inhibits intracellular sodium accumulation via the NHE-1 channel. [3] Sodium does not enter the intracellular space and becomes NCX inactivated. [4] Because of this agent’s mechanism, zinc accumulation is reduced, resulting in reduced neuronal death after global cerebral ischemic insult.

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
