# Peer review of "An Inhibitor of the Sodium–Hydrogen Exchanger-1 (NHE-1), Amiloride, Reduced Zinc Accumulation and Hippocampal Neuronal Death after Ischemia"

_ijms, 2020, doi:10.3390/ijms21124232_

Round 1

Reviewer 1 Report

The study entitled “An inhibitor of the sodium–hydrogen exchanger-1  (NHE-1), Amiloride, Reduces Zinc Accumulation and Hippocampal Neuronal Death after Ischemia” by Beom Seok Kang et al. investigated the protective effects of Amiloride in global cerebral ischemia-reperfusion injury, and its mechanisms involves reducing zinc accumulation and ROS production, as well as reducing astrocytes and glial cells activation. This study has much supporting data. However, I have the following major concerns for this study.

  1. Whether Amiloride is specific inhibitor for NHE-1 should be discussed. It is mentioned that amiloride could regulate several sodium channels in the paper. Since all the conclusion about the mechanism is based on the specificity of this inhibitor, they authors should be careful about their interpretation of the results.
  2. Whether the Amiloride could pass through the blood-brain barrier is not discussed. It is possible that Amiloride may affect other organs for protecting ischemic brains, rather than directly target the ischemic stroke. The authors should be careful in interpretation of those results.
  3. It’s not clear why do the authors choose the dosage of 10 mg/kg.
  4. In the diagram, the authors showed calcium in the signaling cascade, but calcium is not studied in this research. Similarly, this studied dose not study NCX. It seems to me that the authors propose their hypothesis rather than conclusion in the diagram. The authors claimed very complicated signaling, but they didn’t show enough data to support the whole story.
  5. The writing is weak, and it’s hard to get the main points, especially the discussion part. Also, the author tended to repeat the information in their writing.
  6. In the introduction part, the authors discuss the role of superoxide, but they didn’t discuss peroxynitrite, a potent reactive nitrogen species generated from superoxide and nitric oxide. The role of peroxynitrite in mediating zinc activation has also been studied. Recent studies suggest that peroxynitrite plays an important role in BBB damage, brain injury and hemorrhagic transformation. Please discuss.
  7. Zinc has been shown to mediate BBB damage and ischemic brain injury, please discuss this.
  8. There are several studies showing the neuroprotective effects of sodium–hydrogen exchanger inhibitors in stroke models, please discuss.

Author Response

Response to Reviewer 1 comments

Point 1. Whether Amiloride is specific inhibitor for NHE-1 should be discussed. It is mentioned that amiloride could regulate several sodium channels in the paper. Since all the conclusion about the mechanism is based on the specificity of this inhibitor, they authors should be careful about their interpretation of the results.

<Response 1: We agree with this reviewer’s comments that amiloride is a non-specific inhibitor for NHE-1. But it has been used as an NHE-1 inhibitor in several studies [1, 2]. We added this info to the discussion in the line 306-307.>

Point 2. Whether the Amiloride could pass through the blood-brain barrier is not discussed. It is possible that Amiloride may affect other organs for protecting ischemic brains, rather than directly target the ischemic stroke. The authors should be careful in interpretation of those results.

<Response 2: We appreciate this reviewer’s comments. In a previous study, several studies have shown that amiloride can cross the blood-brain barrier and showed protect effects in several brain injuries such as multiple sclerosis, Parkinson’s diseases and cerebral ischemia. We added this sentenceAdditionally, amiloride has been known to cross the blood-brain barrier and has neuroprotective effects against global cerebral ischemia [3, 4]” and referenced it to the line 307-308.>

Point 3. It’s not clear why do the authors choose the dosage of 10 mg/kg.

<Response 3: In a previous study, amiloride was found to have neuroprotective properties using a dose of 10 mg/kg [3, 5, 6]. Thus, we used the same dose of 10 mg/kg. We added this info and reference to the line 309.>

Point 4. In the diagram, the authors showed calcium in the signaling cascade, but calcium is not studied in this research. Similarly, this studied dose not study NCX. It seems to me that the authors propose their hypothesis rather than conclusion in the diagram. The authors claimed very complicated signaling, but they didn’t show enough data to support the whole story.

<Response 4: We appreciate this reviewer’s helpful suggestion. We agree with this reviewer’s concern that we did not study calcium in the present study. We focused on zinc rather than calcium. However, this diagram is a simple hypothesis based on our present study and previous published studies. I hope the reviewer can understand our intention and value in showing this hypothetical model. We added more detailed information in the revised manuscript as follows. Previous studies have shown that NHE-1 inhibitors reduced the activity of NCX [5]. Furthermore, additional studies demonstrated that zinc enters intracellularly through NCX [8, 9]. Thus, we can speculate that NHE-1 inhibition may indirectly inhibit zinc influx into neurons. So, we corrected the diagram and figure 7 legends. And we added the contents and references to the line 299-302.>

Point 5. The writing is weak, and it’s hard to get the main points, especially the discussion part. Also, the author tended to repeat the information in their writing.

<Response 5: We appreciate this reviewer’s comments. We have corrected it in the revised manuscript.>

Point 6. In the introduction part, the authors discuss the role of superoxide, but they didn’t discuss peroxynitrite, a potent reactive nitrogen species generated from superoxide and nitric oxide. The role of peroxynitrite in mediating zinc activation has also been studied. Recent studies suggest that peroxynitrite plays an important role in BBB damage, brain injury and hemorrhagic transformation. Please discuss.

<Response 6: We corrected it. “In addition, several previous studies have suggested that peroxynitrite (PN) toxicity is mediated by intracellular zinc release [10, 11]. PN is produced by a combination of nitric oxide and superoxide. It has been reported that PN are endogenous reactive nitrogen species formed when superoxide radicals, or oxygen, respectively, reacts with nitric oxide formed by iNOS. PN can induce cytoplasmic free zinc release, mitochondria dysfunction, and lead to BBB disruption, and finally neuronal death, in several types of brain injuries [12-14].” We added this info and reference to the line 343-348 in the Discussion part.>

Point 7. Zinc has been shown to mediate BBB damage and ischemic brain injury, please discuss this.

<Response 7: We appreciate this reviewer’s comments and added more details. “Abnormal intracellular zinc accumulation and vesicular zinc release may mediate BBB disruption after brain insults such as ischemia, multiple sclerosis, and traumatic brain injury [15]. Thus, administration of amiloride may decrease zinc accumulation and reduces BBB disruption after GCI.” We added this info and reference to the line 353-356 in discuss part.>

Point 8. There are several studies showing the neuroprotective effects of sodium–hydrogen exchanger inhibitors in stroke models, please discuss.

<Response 8: We appreciate this reviewer’s comments. We added more information and references in the revised manuscript. “Previous studies have demonstrated that amiloride showed neuroprotective effects in two different stroke models; middle cerebral artery occlusion (MCAO) in rat and transient forebrain ischemia in gerbil [27, 37]. However, no studies have been performed with the global cerebral ischemia model with mice. We added this info and reference to the line 272-275.>

Reference

  1. Masereel, B.; Pochet, L.; Laeckmann, D., An overview of inhibitors of Na(+)/H(+) exchanger. Eur J Med Chem 2003, 38, (6), 547-54.
  2. Kleyman, T. R.; Cragoe, E. J., Jr., Amiloride and its analogs as tools in the study of ion transport. J Membr Biol 1988, 105, (1), 1-21.
  3. Durham-Lee, J. C.; Mokkapati, V. U.; Johnson, K. M.; Nesic, O., Amiloride improves locomotor recovery after spinal cord injury. J Neurotrauma 2011, 28, (7), 1319-26.
  4. Xiong, Z. G.; Zhu, X. M.; Chu, X. P.; Minami, M.; Hey, J.; Wei, W. L.; MacDonald, J. F.; Wemmie, J. A.; Price, M. P.; Welsh, M. J.; Simon, R. P., Neuroprotection in ischemia: blocking calcium-permeable acid-sensing ion channels. Cell 2004, 118, (6), 687-98.
  5. Hwang, I. K.; Yoo, K. Y.; An, S. J.; Li, H.; Lee, C. H.; Choi, J. H.; Lee, J. Y.; Lee, B. H.; Kim, Y. M.; Kwon, Y. G.; Won, M. H., Late expression of Na+/H+ exchanger 1 (NHE1) and neuroprotective effects of NHE inhibitor in the gerbil hippocampal CA1 region induced by transient ischemia. Exp Neurol 2008, 212, (2), 314-23.
  6. Arias, R. L.; Sung, M. L.; Vasylyev, D.; Zhang, M. Y.; Albinson, K.; Kubek, K.; Kagan, N.; Beyer, C.; Lin, Q.; Dwyer, J. M.; Zaleska, M. M.; Bowlby, M. R.; Dunlop, J.; Monaghan, M., Amiloride is neuroprotective in an MPTP model of Parkinson's disease. Neurobiol Dis 2008, 31, (3), 334-41.
  7. Shenoda, B., The role of Na+/Ca2+ exchanger subtypes in neuronal ischemic injury. Transl Stroke Res 2015, 6, (3), 181-90.
  8. Sensi, S. L.; Canzoniero, L. M.; Yu, S. P.; Ying, H. S.; Koh, J. Y.; Kerchner, G. A.; Choi, D. W., Measurement of intracellular free zinc in living cortical neurons: routes of entry. J Neurosci 1997, 17, (24), 9554-64.
  9. Sekler, I.; Sensi, S. L.; Hershfinkel, M.; Silverman, W. F., Mechanism and regulation of cellular zinc transport. Mol Med 2007, 13, (7-8), 337-43.
  10. Choi, B. Y.; Jung, J. W.; Suh, S. W., The Emerging Role of Zinc in the Pathogenesis of Multiple Sclerosis. Int J Mol Sci 2017, 18, (10).
  11. Li, S.; Vana, A. C.; Ribeiro, R.; Zhang, Y., Distinct role of nitric oxide and peroxynitrite in mediating oligodendrocyte toxicity in culture and in experimental autoimmune encephalomyelitis. Neuroscience 2011, 184, 107-19.
  12. Zhang, Y.; Wang, H.; Li, J.; Dong, L.; Xu, P.; Chen, W.; Neve, R. L.; Volpe, J. J.; Rosenberg, P. A., Intracellular zinc release and ERK phosphorylation are required upstream of 12-lipoxygenase activation in peroxynitrite toxicity to mature rat oligodendrocytes. J Biol Chem 2006, 281, (14), 9460-70.
  13. Khan, M.; Dhammu, T. S.; Sakakima, H.; Shunmugavel, A.; Gilg, A. G.; Singh, A. K.; Singh, I., The inhibitory effect of S-nitrosoglutathione on blood-brain barrier disruption and peroxynitrite formation in a rat model of experimental stroke. J Neurochem 2012, 123 Suppl 2, 86-97.
  14. Torreilles, F.; Salman-Tabcheh, S.; Guerin, M.; Torreilles, J., Neurodegenerative disorders: the role of peroxynitrite. Brain Res Brain Res Rev 1999, 30, (2), 153-63.
  15. Choi, B. Y.; Lee, S. H.; Choi, H. C.; Lee, S. K.; Yoon, H. S.; Park, J. B.; Chung, W. S.; Suh, S. W., Alcohol dependence treating agent, acamprosate, prevents traumatic brain injury-induced neuron death through vesicular zinc depletion. Transl Res 2019, 207, 1-18.
  16. Li, W.; Ward, R.; Dong, G.; Ergul, A.; O'Connor, P., Neurovascular protection in voltage-gated proton channel Hv1 knock-out rats after ischemic stroke: interaction with Na(+) /H(+) exchanger-1 antagonism. Physiol Rep 2019, 7, (13), e14142.

Reviewer 2 Report

In this manuscript, Kang et al show the effect of NHE-1 inhibition on intracellular zinc accumulation. They used various immunohistological markers to show decreased neuronal loss and improved neuronal survival. Overall the research design is rigorous, and the statistical analyses used are appropriate to answer the research question. Scientifically the study is conducted and represented very well but I recommend getting the manuscript proofread for grammar and incomplete sentences from professional or native English speaker.  Below are some of the examples of that 

line 150: NOS and NO please mention full forms

line 152: GFAP & Iba-1 please mention full forms

line 286: this sentence appears to be incomplete.

line 295: please remove this sentence "Furthermore, we hypothesized...." 

line 302-304: this appears to be an incomplete sentence. 

line 355: this is your conclusion not hypothesize. Please correct it.

line 268-354: throughout the discussion, previous literature is reported using inconsistent tense. It should be corrected to the present tense.

Author Response

Response to Reviewer 2 comments

Point 1. line 150: NOS and NO please mention full forms

<Response 1: We added this. Nitric oxide synthase (NOS) and nitric oxide (NO) in the revised manuscript to the line 152.>

Point 2. line 152: GFAP & Iba-1 please mention full forms

<Response 2: We mentioned it as glial fibrillary acidic protein (GFAP) and ionized calcium binding adaptor molecule 1 (Iba-1) in the revised manuscript at line 154-155. >

Point 3. line 286: this sentence appears to be incomplete.

<Response 3: We appreciate this reviewer's suggestion. We corrected it. “This phenomenon leads to a lack of oxygen and other substrates to the nervous tissue.” We added it to line 292-293.>

Point 4. line 295: please remove this sentence "Furthermore, we hypothesized...."

<Response 4: We removed it.>

Point 5. line 302-304: this appears to be an incomplete sentence.

<Response 5: We appreciate this reviewer’s helpful suggestion. “Following this logic, we hypothesized that GCI-induced hippocampal damage can be protected by reducing intracellular zinc accumulation through inhibition of NHE-1” We revised this sentence at line 310-311.>

Point 6. line 355: this is your conclusion not hypothesize. Please correct it.

<Response 6: We appreciate this reviewer’s helpful suggestion. “Taken together, we concluded that inhibition of NHE-1 by amiloride reduces neuronal death and zinc accumulation.” We revised this sentence at line 365-366.>

Point 7. line 268-354: throughout the discussion, previous literature is reported using inconsistent tense. It should be corrected to the present tense.

<Response 7: We appreciate this reviewer’s helpful suggestion. We corrected the tense.>